# Deletion of the *chd7* Hinders Oligodendrocyte Progenitor Cell Development and Myelination in Zebrafish

**DOI:** 10.3390/ijms241713535

**Published:** 2023-08-31

**Authors:** Lingyu Shi, Zongyi Wang, Yujiao Li, Zheng Song, Wu Yin, Bing Hu

**Affiliations:** 1Center for Advanced Interdisciplinary Science and Biomedicine of IHM, Division of Life Sciences and Medicine, University of Science and Technology of China, Hefei 230026, China; sly135@mail.ustc.edu.cn (L.S.);; 2Division of Life Sciences and Medicine, University of Science and Technology of China, Hefei 230001, China

**Keywords:** *chd7*, oligodendrocyte, myelination, zebrafish

## Abstract

*CHD7*, an encoding ATP-dependent chromodomain helicase DNA-binding protein 7, has been identified as the causative gene involved in CHARGE syndrome (Coloboma of the eye, Heart defects, Atresia choanae, Retardation of growth and/or development, Genital abnormalities and Ear abnormalities). Although studies in rodent models have expanded our understanding of CHD7, its role in oligodendrocyte (OL) differentiation and myelination in zebrafish is still unclear. In this study, we generated a *chd7*-knockout strain with CRISPR/Cas9 in zebrafish. We observed that knockout (KO) of *chd7* intensely impeded the oligodendrocyte progenitor cells’ (OPCs) migration and myelin formation due to massive expression of *chd7* in oilg2^+^ cells, which might provoke upregulation of the MAPK signal pathway. Thus, our study demonstrates that *chd7* is critical to oligodendrocyte migration and myelination during early development in zebrafish and describes a mechanism potentially associated with CHARGE syndrome.

## 1. Introduction

Human chromatin remodeling factor *CHD7*, which encodes an ATP-dependent chromatin helicase DNA-binding protein, CHD7, a member of the SNF2-protein superfamily, is the gene known to be associated to CHARGE syndrome (Coloboma of the eye, Heart defects, Atresia choanae, Retardation of growth and/or development, Genital abnormalities, and Ear abnormalities) [1,2,3]. A proportion of patients exhibited neurological abnormalities, including autistic-like behavior [4,5], anxiety [6], obsessive–compulsive disorder [7], hyperactivity disorder [8], and seizures [9], among others. Notably, the specific mechanisms of these neurological deficits have not been elucidated, and there is no effective pharmacological treatment to improve the associated neurological functions.

In recent years, patients have been reported to have cerebral white matter lesions as a symptom [10,11,12]. As neuronal cell bodies are localized in the gray matter, people have tended to focus more on gray matter when investigating the function of the central nervous system (CNS). However, studies have shown that white matter abnormalities are associated with many neurological disorders [13,14]. Cerebral white matter lesions are typical a demyelinating response of the CNS in the face of various noxious stimuli. This suggests that CHD7 may play a role in the development of myelination in the CNS.

CHD7 forms complexes with different proteins (e.g., BPAF, TRX, and NLK) that bind specifically to the enhancer regions and regulate chromatin conformation, thereby regulating the expression of downstream target genes [15]. Conditional knockdown of *chd7* results in cerebellar hypoplasia [16] and inability of the cochlea to form a hemicircular canal [17]. Previous studies also have shown that heterozygous *chd7* mice exhibit multiple symptoms of CHARGE syndrome, such as ocular defects, impaired vestibular sensation, and inner ear damage. However, currently no homozygous embryos have survived beyond E10.5 [18,19,20]. CHD7 functioned on several stem and precursor cells to promote cell differentiation by activating transcription [21,22]. In mice, CHD7 was shown to act synergistically with SOX10 in myelin development and remyelination [23]. It has also been reported that oligodendrocyte precursor survival and differentiation require chromatin remodeling by Chd7 and Chd8 [24]. In addition, it was found that *chd7* knockdown resulted in a massive reduction of myelinated Schwann cells in zebrafish [25]. However, it is unknown whether *chd7* plays a similar role in oligodendrocyte function in zebrafish.

Zebrafish embryos are translucent and are ideal model organisms for investigating the processes of myelination and remyelination in conjunction with live imaging techniques [26,27,28]. Zebrafish can produce several hundred eggs per week and develop rapidly in vitro, making zebrafish an ideal model for high-throughput experiments [29,30]. The development of myelin is divided into an early formation and an extension phase in the CNSs of zebrafish. Approximately 48 h post-fertilization (hpf), OPCs begin to migrate from the ventral to the dorsal side of the spinal cord and then reach a suitable location to differentiate into OLs, which then gradually form myelin sheaths to encase axons [31,32]. The timing of this process is strictly controlled; otherwise, the quality of the myelin sheath and even the physiological function of the nervous system would be affected [33]. Currently, there are no reports regarding *chd7* in the CNSs of zebrafish. 

In our study, we utilized CRISPR-Cas9 technologies to reveal that *chd7* KO suppressed OPC migration and myelin formation. The hindrance to the OPC migration and myelin formation was possibly due to increased phosphorylation of JNK. Our findings shed light on the potential mechanisms and provided a probable treatment strategy associated with CHARGE syndrome.

## 2. Results

### 2.1. Developmental Expression and Characterization of chd7 in Zebrafish

To explore the temporal and spatial patterns of *chd7* expression patterns, we performed whole-mount in situ hybridization using digoxigenin-labeled *chd7* probes. The results indicated that the zebrafish gene *chd7* was expressed in the whole brain at the stage of 2 dpf and was specifically expressed in the CNS throughout the initiation and extension of myelin development (Figure 1(a–d,a’–d’) and Appendix A). The qPCR results showed that the mRNA expression of *chd7* starts at 6 hpf, decreases gradually thereafter, and maintains a steady level after 2 dpf, which may be related to the fact that *chd7* promotes cell differentiation (Figure 1e). Furthermore, to assess whether *chd7* is expressed in OLs, we divided the Tg (*olig2*: EGFP) strain of zebrafish larvae into olig2^+^ cells and olig2^−^ cells via flow sorting as the OLs were labeled by EGFP. The results showed that *chd7* was highly expressed in oilg2^+^ cells (Figure 1f,g). This evidence suggested that *chd7* is commonly expressed in the CNS and highly expressed in oilg2^+^ cells during myelin development, implying that *chd7* might play an important role in myelination. 

### 2.2. Construction of chd7 Homozygous Mutant Zebrafish

To obtain KO strains, we designed a target site in the first exon of the *chd7* gene and microinjected small guide RNA (sgRNA) along with recombinant cas9 protein at the one-cell stage (Figure 2a). After three generations of breeding, we obtained homozygous chd7 mutants (Figure 2b). Sequencing results in the F2 generation zebrafish revealed a 16-base pair deletion in exon 1 of *chd7* that causes a frameshift mutation (Figure 2c,d). This mutation causes a premature stop codon and truncated CHD7 protein (Figure 2e). These results confirmed the successful construction of *chd7* homozygous mutant zebrafish strains and showed that the mutation is stably inherited.

### 2.3. Knockout of chd7 Caused Hyperactivity and Malformation 

We used spontaneous movement to detect whether zebrafish motility was altered. We counted the cumulative distance of spontaneous movements of 6 dpf zebrafish larvae within one hour and found that the cumulative distance was significantly increased in the KO group compared to the wild-type (WT) group (Figure 3a,b), indicating that *chd7* KO leads to a malfunction of locomotor function. In addition, qPCR results showed that the expression of motor neuron marker genes *isl1* and *isl2a* were significantly upregulated in the KO group (Figure 3c). Subsequently, we observed morphological phenotypes such as microcephaly, ocular malformations, pericardial edema, and spinal curvature in the KO group (Figure 3d). These phenotypes were also consistent with the clinical signs of CHARGE syndrome, confirming the validity of our model.

### 2.4. Knockout of chd7 Slowed down the Migration of OL to the Dorsal Side

To specifically confirm that the reduced migration of OPCs is caused by a functional loss of *chd7*, we successfully established *chd7* KO strains by removing 16 bp in the first exon of the gene *chd7*. We observed that at 3, 4, and 5 dpf, the number of olig2^+^ cells that migrated dorsally was significantly reduced in the *chd7^−/−^* group compared to the *chd7^+/+^* group, and the difference was diminished at 6 dpf (Figure 4a,b). Interestingly, we found no significant difference in the number of total oilg2^+^ cells between the two groups (Figure 4c,d). These results suggested that loss of CHD7 protein causes delayed OPC migration, but eventually a total migration similar to that of *chd7^+/+^* can be achieved.

### 2.5. Chd7 Knockout Hindered Myelination of OL

The process of myelination of zebrafish OLs starts at the head and then gradually goes to the tail [34]. Moreover, the axons of Mauthner cells are the thickest axons in the zebrafish spinal cord, and their axons are the first to be myelinated [35]. Then, utilizing this time series, we used the number of somites corresponding to the location of myelin-coated Mauthner cell axons as a criterion to assess the process of myelination. We identified that at 70 hpf, the *chd7^+/+^* group had wrapped about 9 somites in length, while the KO group had wrapped only about 2 somites. At 75 hpf and 80 hpf, the length of myelin wrapping in the KO group was also much less than that in the WT group (Figure 5a,b). To further examine the variations in myelination, we randomly labeled individual mature OLs with plasmid UAS-*mbp*-EGFP-CAAX and observed that the number of individual OL-wrapped myelin segments in the *chd7^−/−^* group was approximately 45% smaller than the *chd7^+/+^* group (Figure 5c,d).

The gene *olig2* regulates the maturation and differentiation of OPCs, while *mbp* is a marker gene for myelination. Our qPCR results showed that both *olig2* and *mbp* mRNA levels were significantly reduced after *chd7* KO (Figure 5e). The protein expression of MBP was also substantially lower in the KO group as compared to the *chd7^+/+^* group (Figure 5f). 

To further investigate the quality of myelination, we observed the myelination of neurons in the spinal cord of the control (*chd7^+/+^*) and KO (*chd7^−/−^*) groups at 5 dpf using transmission electron microscopy. From the electron micrographs, we could observe that the number of myelinated axons was less in the KO group compared to the control group (Figure 6b). In addition, we found that many myelinated axons in the KO group had a tendency to demyelinate or produce swollen myelin sheaths (Figure 6a,b). Therefore, we measured the thickness of myelin wrapping by dividing the circumference of the innermost myelin layer of myelinated neurons by the circumference of the outermost myelin wrapping. In other words, the larger this ratio is, the thinner the myelin sheath is. The results showed that the ratio of myelin in the KO group of Mautner cells and other ventral axons was larger than that of the control group, i.e., the myelin was thinner, indicating immature myelination and possible partial loss of function (Figure 6c,d). Thus, these results together suggested that the deletion of CHD7 protein can significantly hinder myelin formation.

### 2.6. CHD7 Might Affect Myelin Development through MAPK Signaling Pathway

Next, we performed bioinformatics analyses, including GO, GSEA, and pathway analysis, by utilizing transcriptome sequencing data, which were reported on in the previous study [9]. We discovered that the significantly dysregulated gene expression was mainly distributed in the MAPK signaling pathway, cell adhesion, calcium signaling pathway, and lipid transport (Figure 7a–d). Among them, the MAPK signaling pathway attracted our attention most because it plays a significant role in a variety of neurodevelopmental disorders. Previous studies have shown that c-Jun is a negative regulator of myelination [36], LPS-sensitized HI causes white matter injury via JNK activation [37], and FGF21 can act as a negative regulator for the myelin development process via activation of p38 MAPK/c-Jun [38]. Combined with the analysis results of the ChIP-seq dataset, we found that many dysregulated genes in the MAPK pathway are direct targets of CHD7 in mouse and human cell lines. Thus, we examined the mRNA levels of these target genes via qPCR, and the results showed that they were significantly up- or down-regulated in the KO group, consistent with the previous transcriptome sequencing data [9] (Figure 7e). In addition, we identified a significant increase in phosphorylated JNK protein levels in the KO group compared to WT (Figure 7f).

The above results together demonstrated that knockout of *chd7* promotes the phosphorylation of JNK protein and has the potential to regulate myelin development via the MAPK pathway.

## 3. Discussion

Since CHARGE syndrome was first clinically described [39], considerable progress has been made in the pathology and genetics of CHARGE syndrome, yet the mechanisms by which patients develop white matter defects have not been clarified. In this study, we showed for the first time, as far as we know, that a functional deficiency of *chd7* delayed the migration and myelin development of OLs in zebrafish. Moreover, we found that *chd7* KO leads to morphological and behavioral disorders. Combining transcriptomics and chip-seq databases, we also found that these phenotypes might be aroused via the JNK/MAPK signaling pathway.

Using in situ hybridization and flow sorting of the Tg (*olig2*: EGFP) strain of zebrafish, we observed that *chd7* was stably expressed in the CNS and was enriched in OLs during the developmental stages of zebrafish. This result was consistent with that obtained from mouse models [23]. Importantly, we observed a significant reduction in the number of OPCs migrating dorsally in the KO group. Interestingly, this difference disappeared at 6 dpf, suggesting that *chd7* deletion led to only the initial decrement of OPCs migration. We also observed that in the *chd7^−/−^* group, the onset of myelination was delayed and the rate of myelination was reduced. This might be due to an altered ability of OL differentiation. Moreover, using the UAS-*mbp*-EGFP-CAAX randomly labeled plasmid, we found that *chd7* KO results in fewer myelinated branches formed by individual OL, suggesting that the myelination potential of OL was indeed severely disrupted. Increasing evidence of white matter abnormalities has been found in various neurological diseases, such as Huntington’s disease, multiple sclerosis, and Alzheimer’s disease, but studies of the mechanisms remain unclear. OPCs are meant to migrate, differentiate, and eventually form intact, functionally mature myelin sheaths at the correct time points. The mechanism of myelination is not simply wrapping neuronal axons; during this process, neurons, astrocytes, microglia, and OLs closely communicate and interact with each other [40,41,42,43]. Therefore, future studies of myelin abnormalities in neurological diseases could take cellular communication into consideration.

There have been several other animal models used to study the function of *chd7*, including *CHD7* regulation of bone–fat balance in mice [44], kismet (orthologue of chd7 in Drosophila) promotion of transcription elongation [45], and the role of *CHD7* in the formation of the multipotent migratory neural crest in Xenopus [46]. While homozygote mice die in utero [47,48], *chd7^−/−^* zebrafish can survive stably up to 10 dpf, probably due to the difference between oviposition and fecundity. Zebrafish embryos are incubated in vitro, shed their egg membranes at about 2 dpf, and derive most of their nutrition from the yolk until 5 dpf [49]. In contrast, mouse embryos require maturation within the mother’s body for direct output, which generally takes about 22 days [50]. During this process, KO of *chd7* hinders cell differentiation, organ development, and angiogenesis [51]. Symptoms of esophageal malformations have recently been reported in patients with CHARGE syndrome [10]. Feeding difficulties, then, are also a major reason for the low survival rate of neonatal patients. With these functions blocked, fetal death in mice seems to be the inevitable outcome. On the other hand, this study also demonstrated that the zebrafish model which mimics CHARGE syndrome may have advantages above the mouse model.

We also used bioinformatics to find that JNK phosphorylation is greatly increased after *chd7* functional deletion and its downstream gene expression is differentially altered, suggesting that *chd7* may regulate myelin regulation via the MAPK signaling pathway. Reasonably, other pathways may also be involved in regulating the functional direction of OLs, which requires further investigation.

In summary, we have shown for the first time that *chd7* regulates the migration and early differentiation of OLs in the zebrafish CNS, and we identified the expression pattern of *chd7* in early myelin development. This demonstrates that zebrafish can serve as an excellent model for studying myelin-related diseases and provide new insights into clinical treatment of CHARGE syndrome.

## 4. Methods and Materials

### 4.1. Zebrafish Strains and Maintenance

Zebrafish were cultured under laboratory conditions, maintained at 28.5 °C and a 14/10 h light/dark cycle [52]. Zebrafish embryos were collected after natural spawning and staged by hpf and days post-fertilization (dpf) according to established criteria [53]. To achieve in vivo staging, larvae were raised in fetal medium containing 0.2 mM N-phenylthiourea (Sigma-Aldrich, St. Louis, MO, USA) to inhibit pigmentation. All animal manipulations described in this study were performed in strict accordance with the guidelines and regulations established by the University of Science and Technology Animal Resource Center (USTC) and the University Animal Care and Use Committee. All protocols were approved by the University of Science and Technology Animal Experimentation Ethics Committee (license number: USTCACUC1103013).

The following zebrafish lines were used in this study: wild-type (WT), Tg (*olig2*: EGFP) [54], and Tg (*mbp*: EGFP-CAAX) which were kindly provided by Prof. David A. Lyons, University of Edinburgh, UK [35].

### 4.2. Genome Editing

Exon 1 of the *chd7* gene was designated as the Cas9 target. At the one-cell stage, Cas9 protein was co-injected along with chd7 guide sgRNA into zebrafish embryos to obtain the F0 generation. The F0 generation was raised to adulthood and crossed with AB wild-type fish, after which the offspring were screened for the same sense mutation and continued to be bred (F1). F1 self-crosses yielded *chd7*^−/−^, *chd7^+/−^* and *chd7^+/+^* (F2), Tg (*olig2*: EGFP), *chd7^−/−^*, and Tg (*mbp*: EGFP-CAAX); *chd7^−/−^* zebrafish were bred using the same procedure.

### 4.3. Quantitative Real-Time PCR

Quantitative RT-PCR was performed using corresponding primers (Table 1). The final reaction volume was 20 μL, containing 10 μL SYBR premix (Takara, Kyoto, Japan). β-actin [33] was used as the internal control.

### 4.4. Western Blot

To detect protein expression of JNK and p-JNK in zebrafish larvae, we collected equal numbers of WT and *chd7* KO larvae at 3 dpf. Larvae were lysed with RIPA buffer (Servicebio, Wuhan, China). Ten μL of extract per well was applied to the SDS-PAGE gel. Western blots were performed using the antibodies to p-JNK1/2/3 (rabbit; 1:1000; Abmart, Shanghai, China; Cat# TA3320S), JNK1/2/3 (rabbit; 1:1000; Abmart; Cat# T55490S), and GAPDH (rabbit;1:2000; HUABIO, Hangzhou, China; Cat# ET1601-4) as primary antibodies. Secondary anti-rabbit antibodies (goat; Proteintech, Chicago, IL, USA; Cat# SA00001-2) were diluted to 1:5000 at room temperature.

### 4.5. In Vivo Imaging

Zebrafish larvae were anesthetized with 0.4% MS222 (Sigma) and then embedded in 1% low-melting agarose. Larvae were scanned with a FV1000 confocal microscope (Olympus, Tokyo, Japan) under a 10× objective lens or a 40× objective lens. In vivo imaging of OPC migration and myelination were conducted as described previously [55].

### 4.6. Whole-Mount In Situ Hybridization

To generate a probe for *chd7*, a 543 bp fragment of the *chd7* gene with a T7 promoter added was amplified from 24 hpf zebrafish cDNA. The fragment was used as template for synthesizing an antisense *chd7* RNA probe labeled with digoxigenin (Roche, Mannheim, Germany) with in vitro transcription systems (Invitrogen, Waltham, MA, USA).

### 4.7. Fluorescence-Activated Cell Sorting

At 4 dpf, three hundred Tg (*olig2*: EGFP) zebrafish larvae were selected and washed twice with phosphate-buffered saline. The embryos were chopped and digested in 10 mL of trypsin for 20 min. After cell screening, single cell suspensions were obtained. Samples were analyzed via flow cytometry (BD FACS Aria III, Franklin Lakes, NJ, USA) and divided into EGFP^+^ cells and EGFP^−^ cells. Total RNA was extracted from the sorted cells using Trizol (Invitrogen, Waltham, MA, USA).

### 4.8. Spontaneous Movement Assay

Six-dpf zebrafish larvae were placed in a 24-well plate on a platform. One hour of free swimming in the dark was recorded and analyzed with Viewpoint equipment (Viewpoint, Lyon, France) and Zebralab software (Zebralab version 5.13.0.270). The cumulative swimming distance was statistically used to determine whether the zebrafish’s motility was altered, as it had previously been reported that *chd7* heterozygous mice and 5 dpf *chd7^−/−^* zebrafish exhibited hyperactivity and the later showed particularly prominent hyperactivity in the dark cycle [9,56].

### 4.9. Transmission Electron Microscopy Imaging

MS-222 lethal working solution and glutaraldehyde fixative were freshly prepared using PBS; the larvae were anesthetized and lethally transferred into 2.5% glutaraldehyde (Sangon, Shanghai, China) quickly and fixed overnight at room temperature on a shaker. The fixed fish were washed three times with 0.1 M Cacodylate buffer for 5 min each time, then fixed in 1% osmium acid for 2 h at room temperature and washed once with ddH_2_O. Gradient dehydration was performed using different concentrations of ethanol into 50% ethanol for 15 min before being placed in 70% uranyl acetate ethanol solution on a shaker overnight. Dehydration was continued using 80%, 95%, and 100% ethanol for 15 min each. The dehydrated larvae were treated using a gradient of propylene oxide/epoxy resin for approximately 1 h each time and finally embedded in epoxy resin. After orientation using pointed forceps, they were placed in an oven at 65 °C for overnight curing. The cured resin blocks were briefly trimmed with a razor blade, placed in an LKB NOVA ultra-thin sectioning machine (Lecia, Kista, Sweden) for ultra-thin sectioning (70–90 nm), affixed to copper mesh, and imaged using a JEM-1230 electron microscope (JEOL, Tokyo, Japan), and the remaining resin blocks were placed back into 2 mL centrifugal tubes and sealed in a drying box for storage. The acquired negatives were scanned with a scanner and analyzed and processed using ImageJ software (ImageJ version 8).

In this study, 5 Mauthner cells and 102 myelinated axons cells of *chd7^+/+^* group were measured while 6 Mauthner cells and 53 myelinated axons cells of *chd7^−/−^* group were measured.

### 4.10. Statistical Analysis

Statistical data comparisons were performed using Student’s t-test or one-way analysis of variance (ANOVA) using GraphPad Prism version 8.0.1 (GraphPad Software, San Diego, CA, USA, www.graphpad.com, accessed on 5 December, 2020). Values were presented as means ± s.e.m.s. Sample sizes similar to those of previous work were used [57,58,59]. For qRT-PCR analyses, data were obtained from at least three repeated experiments using three independent samples. A value of *p* < 0.05 was considered significant (* *p* < 0.05, ** *p* < 0.01, *** *p* < 0.001 and **** *p* < 0.0001).

## Figures and Tables

**Figure 1 ijms-24-13535-f001:**
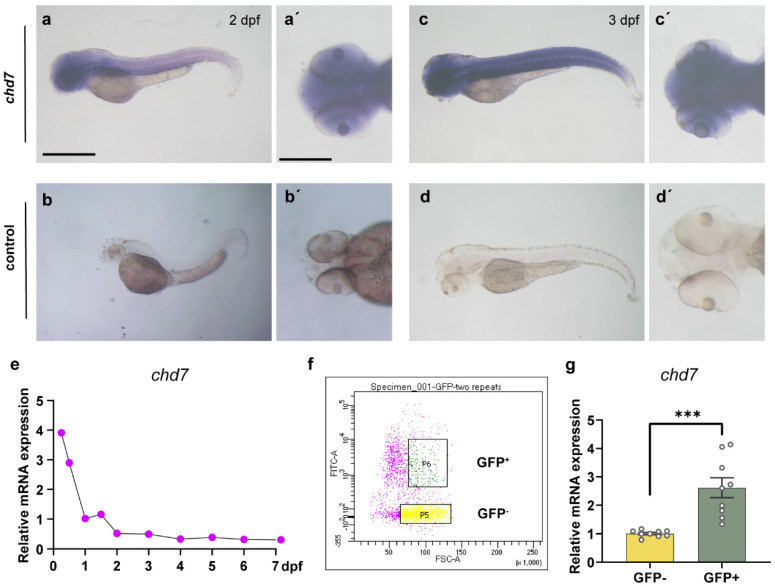
The expression pattern of *chd7* at the early stage. (**a**–**d**) Zebrafish at 2–3 dpf using WISH displaying high expression of *chd7* in the brain, eyes and spinal cord (lateral view). (**a’**–**d’**) Dorsal view of the brain. All embryos shown in both the lateral and dorsal view. Scale bar in (**a**–**d**), 10 μm; Scale bar in (**a’**–**d’**), 5 μm. (**e**) Q-PCR results showed that *chd7* mRNA expression begins at 6 hpf and decreases gradually thereafter. (**f**) Representative image of FACS analysis of olig2^+^ cells and olig2^−^ cells from the Tg (*olig2*: EGFP) line. (**g**) Real-time Q-PCR results of olig2^+^ cells and olig2^−^ cells. Data shown as mean ± sem. *** *p* = 0.0003, unpaired Student’s two-tailed *t*-test.

**Figure 2 ijms-24-13535-f002:**
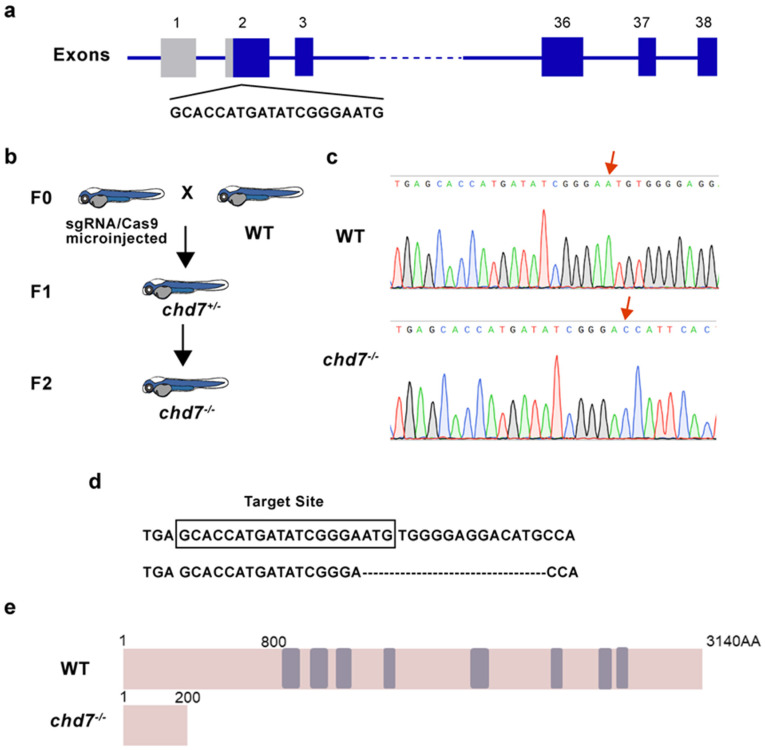
Identification of chd7 knockout zebrafish. (**a**) Schematic of the Cas9-sgRNA targeted site located in the first exon of *chd7*. (**b**) After crossbreeding and self-crossing, homozygous mutant strains were obtained. F0: sgRNA/Cas9 microinjected zebrafish was mated with WT zebrafish; F1: the heterozygous descendant of F0, which have the same type of mutation; F2: homozygous descendant of F1. (**c**,**d**) Sequencing results showed that 16 bp were deleted in the *chd7^−/−^* zebrafish. (**e**) Schematic of the protein sequence showed that CHD7 mutated zebrafish expressed truncated CHD7 composed of 200 amino acids, while wild-type CHD7 is composed of 3140 amino acids.

**Figure 3 ijms-24-13535-f003:**
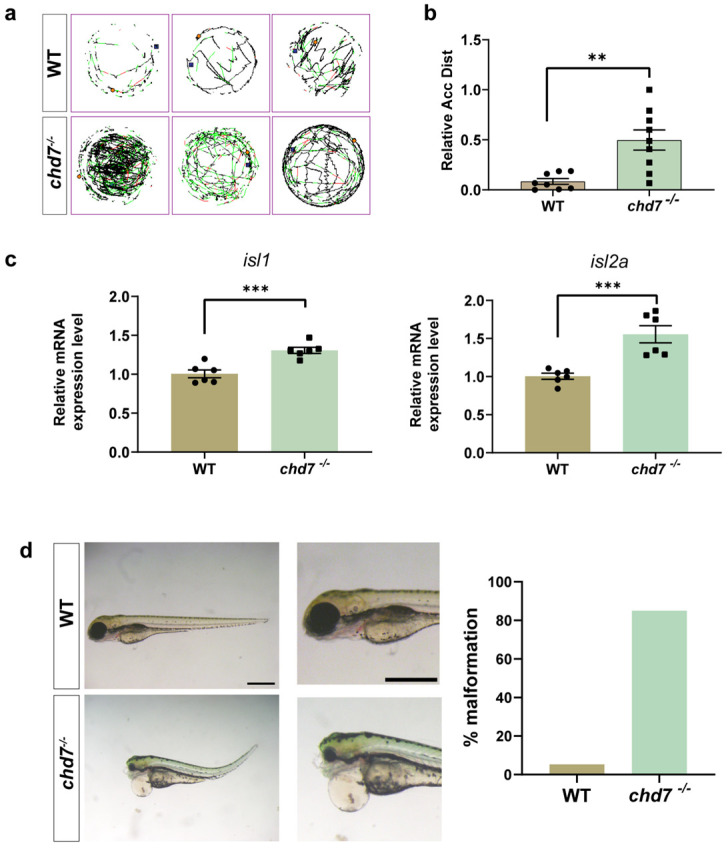
Knockout of *chd7* caused developmental behavioral defects. (**a**,**b**) The line illustrates the 6 dpf zebrafish larvae’s swimming trajectory differences from the WT and *chd7*^−/−^ groups evaluated over 1 h. ** *p* = 0.0021, unpaired Student’s two-tailed *t*-test. (**c**) Real-time Q-PCR results of motor neuron marker genes *isl1* (*** *p* = 0.0007) and *isl2a* (*** *p* = 0.001). Data were shown as mean ± sem. unpaired Student’s two-tailed *t*-test. (**d**) Representative morphological image of embryos and analysis of malformation rate from the WT and *chd7^−/−^* groups at 3 dpf. Scale bar: 100 μm.

**Figure 4 ijms-24-13535-f004:**
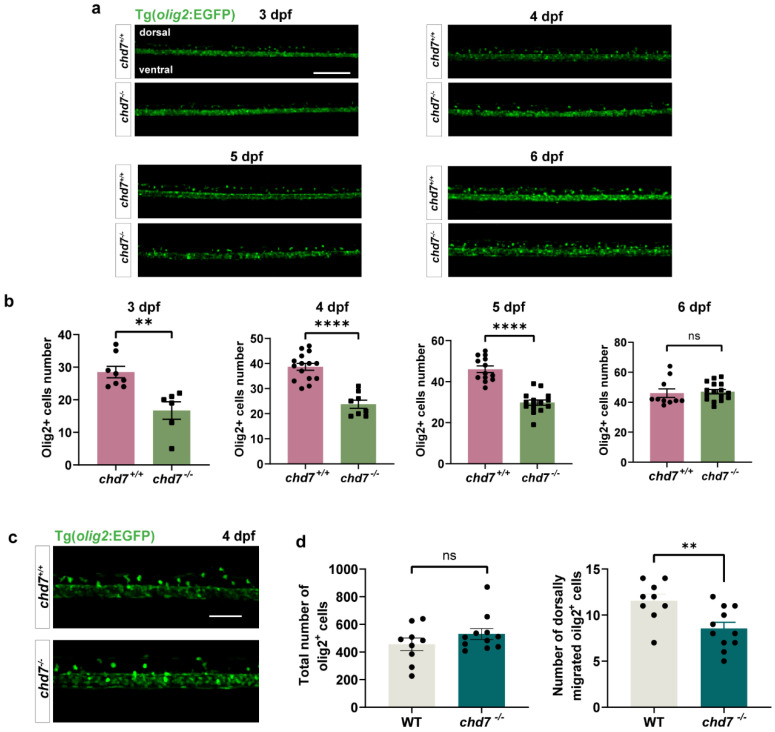
*Chd7* knockout suppressed OPC migration. (**a**) Representative image of dorsally migrated olig2^+^ cells at 3, 4, 5, and 6 dpf between *chd7^+/+^* and *chd7^−/−^* groups. Scale bar: 100 μm. (**b**) The number of dorsally migrated olig2^+^ cells at 3, 4, 5, and 6 dpf decreased in *chd7^−/−^* group. Data shown as mean ± sem. 3 dpf, ** *p* = 0.0023; 4 dpf, **** *p* < 0.0001; 5 dpf, **** *p* < 0.0001; 6 dpf, *p* = 0.7349, ns: not significant, unpaired Student’s two-tailed *t*-test. (**c**) Representative image of total and dorsally migrated olig2^+^ cells at 4 dpf between *chd7^+/+^* and *chd7^−/−^* group. Scale bar: 10 μm. (**d**) The number of total olig2^+^ cells at 4 dpf did not differ significantly between the two groups, ns: not significant. The number of dorsally migrated olig2^+^ cells at 4 dpf decreased in *chd7^−/−^* group. ** *p* = 0.0075, unpaired Student’s two-tailed *t*-test.

**Figure 5 ijms-24-13535-f005:**
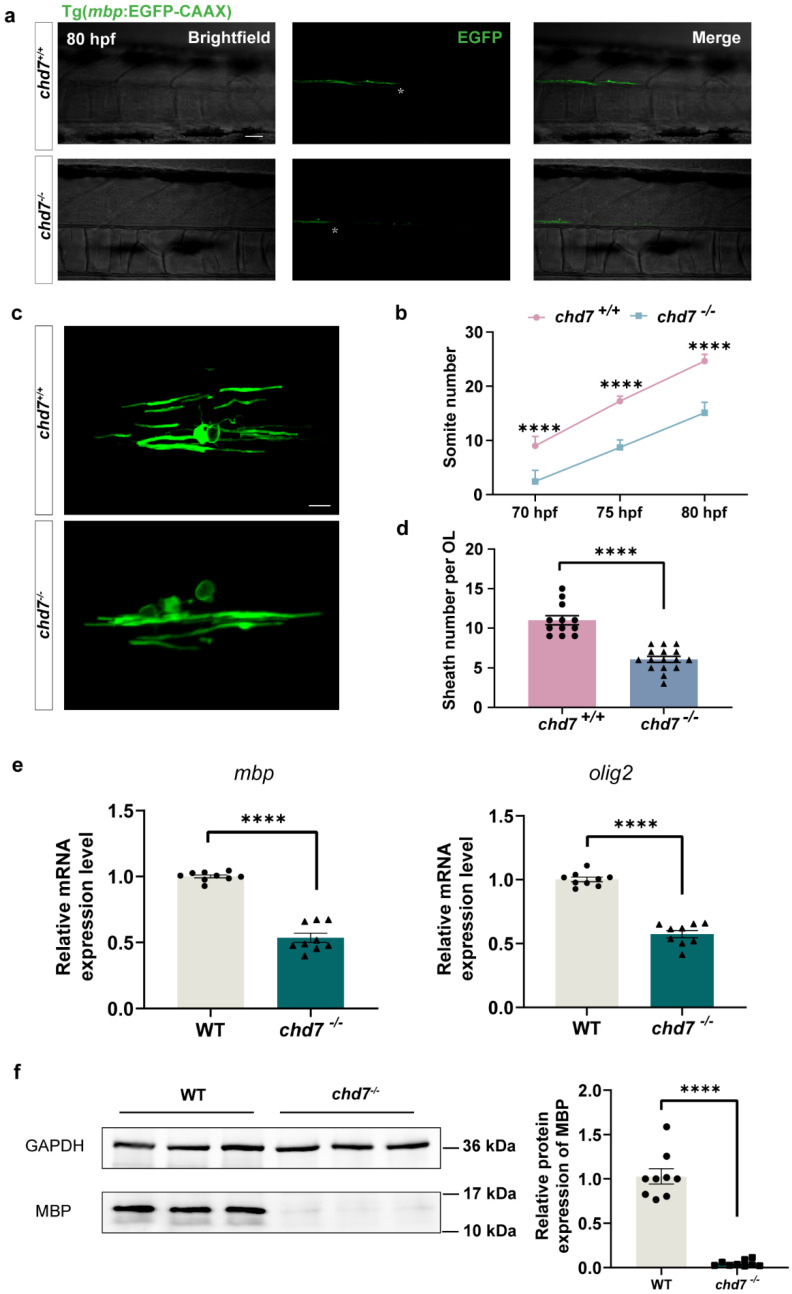
*Chd7* knockout mutant affected the myelin potential of OL. (**a**) The most posterior myelin sheath in the spinal cord at 80 hpf during myelination of the Mauthner axon between *chd7^+/+^* and *chd7^−/−^* groups. Asterisks represent the terminals of myelin-coated axons. Scale bars:100 μm. (**b**) The most posterior myelin sheath at 70 hpf was somite 9 in the wild-type and somite 2 in mutant groups; at 75 hpf was somite 17 in the wild-type and somite 9 in mutant groups; and at 80 hpf was somite 25 in the wild-type and somite 15 in mutant groups. (**c**) Typical confocal imaging of individual oligodendrocytes labeled with plasmid mbp-EGFP-CAAX in the wild-type and mutant groups. Scale bar: 50 μm. (**d**) Number of individual oligodendrocyte-wrapped myelin segments in the wild-type and mutant groups. Wild type: 12 cells derived from 10 juveniles; mutant group: 16 cells derived from 9 juveniles. (**e**) Real-time Q-PCR results of gene *olig2* and *mbp*. (**f**) Protein MBP was decreased in knockout group as shown by Western blot quantification. Data were shown as mean ± sem. **** *p* < 0.0001, unpaired Student’s two-tailed *t*-test.

**Figure 6 ijms-24-13535-f006:**
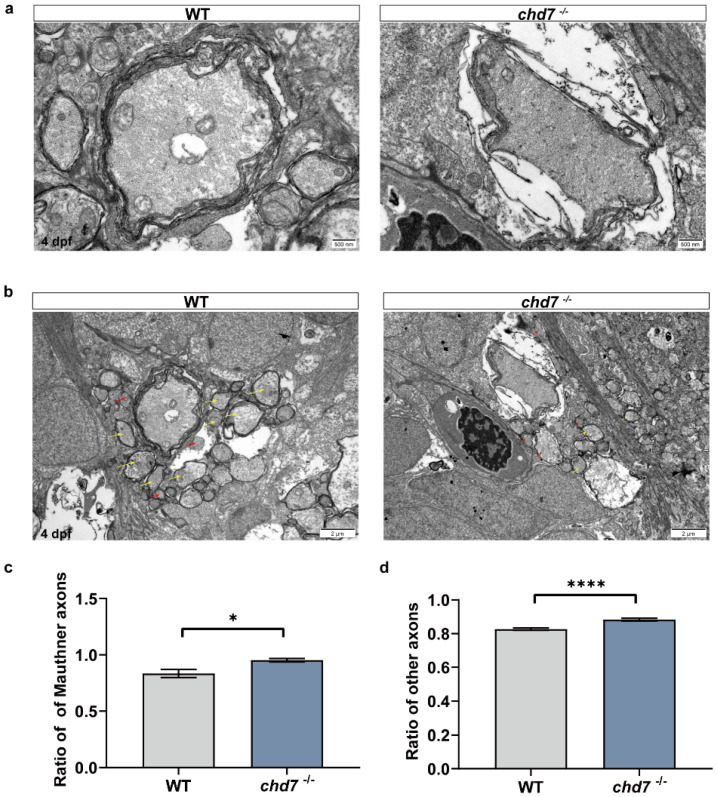
Deficiency of *chd7* lowered the quality of myelin sheath. (**a**) Transmission electron micrograph of the axons of Mauthner cells in the WT and KO groups of 5 dpf zebrafish larvae. Scale bars: 500 nm. (**b**) Transmission electron micrograph of other ventral axons in the WT and KO groups of 5 dpf zebrafish larvae, with yellow arrows indicating myelinated axons and red arrows indicating unmyelinated axons. Scale bars: 2 μm. (**c**,**d**) Myelin thickness analysis of Mauthner cells and other ventral axons. Compared with the WT group, both Mauthner cells and other ventral axons in the KO group had a higher ratio, implying thinner myelin sheaths. Data shown as mean ± sem. unpaired Student’s two-tailed *t*-test, * *p* < 0.05, **** *p* < 0.0001.

**Figure 7 ijms-24-13535-f007:**
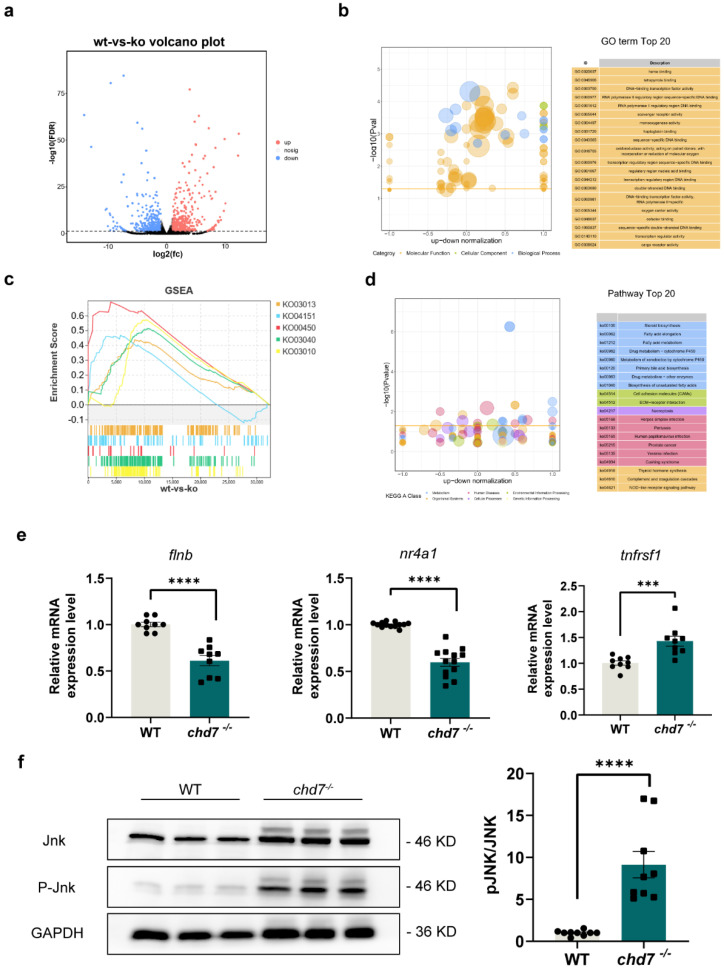
*Chd7* may regulate myelin development via the MAPK pathway. (**a**) Volcano plot showing each gene plotted according to its log2 fold change. All highly differentially expressed genes with *p* < 0.05 are in orange with fold change > 1.5. In total, 589 genes were significantly up-regulated and 438 genes were significantly down-regulated. (**b**) Gene ontology showing the 20 most significant functional classifications for differential gene enrichment. (**c**) Gene set enrichment analysis determining the significantly down-regulated pathway activity in the *chd7* knockout group. (**d**) Kyoto Encyclopedia of Genes and Genomes analysis determining the top 20 enriched pathways of the differentially expressed genes. (**e**) Real-time Q-PCR results showed that the target genes of protein CHD7 are significantly changed in expression after *chd7* KO. (**f**) JNK activation via phosphorylation was increased in knockout group as shown via Western blot quantification. Data shown as mean ± sem. unpaired Student’s two-tailed *t*-test, *** *p* = 0.001, **** *p* < 0.0001.

**Table 1 ijms-24-13535-t001:** Plasmid constructs primer sequences.

*chd7*-sgRNA-F	TAATACGACTCACTACACCATGATATCGGGAATGGT TTTAGAGCTAGAAATAGC
*chd7*-sgRNA-R	AGCACCGACTCGGTGCCACT
*chd7*-F	ACCCAGGCATGATGAGCCTCTT
*chd7*-R	CAGGCACCTGCATTGGTTGAGCA
*β-actin*-qPCR-F	CCCTGTTCCAGCCATCCTT
*β-actin*-qPCR-R	TTGAAAGTGGTCTCGTGGATACC
*chd7*-qPCR-F	AGGGACACACAGGGACGTAT
*chd7*-qPCR-R	CGCCATGGCACATTTCGTAG
*mbp*-qPCR-R	CGAGGAGAGGACACAAAGC
*mbp*-qPCR-F	CCGTCGTGGAGACGTCAA
*olig2*-qPCR-F	ATCCGTCCAGTTGTGGCACT
*olig2*-qPCR-R	TGGTGGAAGCAGAGGATGGT
*chd7*-situ-F	TAATACGACTCACTATAGACGGTGTTGGAAGGAACTTG
*chd7*-situ-R	ACGGTGTTGGAAGGAACTTG
*flnb*-qPCR-F	GTGGACCCTGATACAGCATTAC
*flnb*-qPCR-R	ATCCCAGAAGCCTCATCTCT
*tnfrsf1a*-qPCR-F	AGACAGTGCAAATGCCAGGA
*tnfrsf1a*-qPCR-R	TCCTGGCTGACATTTATCGCA
*nr4a1*-qPCR-F	AGGCCATGTCGATCTGTGATT
*nr4a1*-qPCR-R	GTGCGTACGAAGCTTTGGTG

## Data Availability

All data generated or analyzed during this study are included in this published article and its Appendix A.

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
