# Peer review of "Deletion of the chd7 Hinders Oligodendrocyte Progenitor Cell Development and Myelination in Zebrafish"

_ijms, 2023, doi:10.3390/ijms241713535_

Round 1

Reviewer 1 Report

The authors generated a chd7-knockout strain with CRISPR/Cas9 in zebrafish and performed immunohistochemistry, immunoblotting, and bioinformatics. The authors showed that the zebrafish lacking chd7 intensely impeded the OPC migration and myelin formation. It is an interesting finding, but there are several concerns.   As the authors made a global chd7 ko zebrafish, the interpretation of data needs more caution. For example, the authors showed that Motor neuron marker genes, isl1, and isl2a, are unregulated but did not examine if the number of motor neurons changed.   The g-ratio analysis is not convincing to me because the myelin at 4dpf is thin even in the wild type, and the myelin sheath was not compact and destructed their structure. In addition, the higher g-ratio implies thinner myelin sheaths or axons become larger in chd7ko. The authors should examine if there is any change in the axon diameter since chd7-/- zebrafish is a global KO.    The authors showed that the reduction of myelin and Jnk1 was upregulated in chd7-/-. However, Lorenzati et al.(2021) showed that the global ablation of the JNK1 results in a significant reduction of myelin in the cerebral cortex and corpus callosum at both postnatal and adult stages. Myelin alterations are accompanied by higher OPC density and proliferation during the first weeks of life, consistent with a transient alteration of mechanisms regulating OPC self-renewal and differentiation. Their data are opposite from the author's group. How do you explain the discrepancy?   Lorenzati M, Boda E, Parolisi R, Bonato M, Borsello T, Herdegen T, Buffo A, Vercelli A. c-Jun N-terminal kinase 1 (JNK1) modulates oligodendrocyte progenitor cell architecture, proliferation and myelination. Sci Rep. 2021 Mar 31;11(1):7264. doi: 10.1038/s41598-021-86673-6. PMID: 33790350; PMCID: PMC8012703.   The explanation of figure 7a,b,c,d is not sufficient. The authors should provide more information to understand the figure in the main text. The font size is too small to read.   In the abstract, “We observed that knockout (KO) of chd7 intensely impeded the oligodendrocyte progenitor cells (OPCs) migration and myelin formation on account of massive expression of chd7 in OPCs, which might evoke upregulation of the MAPK signal pathway.” The authors made a global chd7-/- zebrafish. How "massive expression of chd7 in OPCs” happened?   Minor issues: The supplemental data figure 2 is important data and should be in the main figure.   EM procedure in the method section is missing. How many myelinated axons were measured?   Marie et al.(2018) examined that chromatin remodeling by Chd7 and Chd8 requires OPC survival and differentiation. It should be included in the manuscript. Marie C, Clavairoly A, Frah M, Hmidan H, Yan J, Zhao C, Van Steenwinckel J, Daveau R, Zalc B, Hassan B, Thomas JL, Gressens P, Ravassard P, Moszer I, Martin DM, Lu QR, Parras C. Oligodendrocyte precursor survival and differentiation requires chromatin remodeling by Chd7 and Chd8. Proc Natl Acad Sci U S A. 2018 Aug 28;115(35):E8246-E8255. doi: 10.1073/pnas.1802620115. Epub 2018 Aug 14. PMID: 30108144; PMCID: PMC6126750.

Author Response

Reviewer #1: The authors generated a chd7-knockout strain with CRISPR/Cas9 in zebrafish and performed immunohistochemistry, immunoblotting, and bioinformatics. The authors showed that the zebrafish lacking chd7 intensely impeded the OPC migration and myelin formation. It is an interesting finding, but there are several concerns.  

Answer:

We are sincerely grateful for the constructive comments on our manuscript and for giving us the opportunity to revise this manuscript. We deeply appreciate the recognition that we generated a new chd7-/- mutant line, in which damaged morphology and behavior, reduced migration of OPCs (oligodendrocyte precursor cells), and hindered myelination during the early developmental stages were observed, and we used a wide variety of different established methods, such as CRISPR/Cas9, microinjection, confocal imaging in vivo, transmission electron microscopy, and other standard methods. As suggested by the reviewer, we incorporated the suggestions and comments into the revision.

We hope the revised manuscript is suitable for possible publication.

Point 1: As the authors made a global chd7 ko zebrafish, the interpretation of data needs more caution. For example, the authors showed that Motor neuron marker genes, isl1, and isl2a, are unregulated but did not examine if the number of motor neurons changed.  

Response 1: We thank the reviewer for the valuable comments. We found that the chd7-/- group swam longer distances for the same durations compared to the control group, after which the expression of motor neuron marker genes was measured. The upregulated isl1 and isl2a were used as additional index that chd7 knockout causes hyperactivity in zebrafish. We did not count all the motor neurons as: 1) we mainly focused on the myelin, oligodendrocytes, and OPCs in this study; 2) the major motor neurons are the Mauthner cells which did not show remarkable changes.

Point 2: The g-ratio analysis is not convincing to me because the myelin at 4 dpf is thin even in the wild type, and the myelin sheath was not compact and destructed their structure. In addition, the higher g-ratio implies thinner myelin sheaths or axons become larger in chd7 ko. The authors should examine if there is any change in the axon diameter since chd7-/- zebrafish is a global KO.

Response 2: We thank the reviewer for the thoughtful comments. Considering that the myelin sheath package is not intact during the early stages of zebrafish development, we actually measured the thickness of myelin on 5 dpf in this study. Notably, we found that the chd7-/- group showed significant swelling and demyelination, which resulted in the inability to characterize the thickness of the myelin sheaths by g-ratio. Therefore, we decided to count the thickness of myelin by dividing the circumference of the innermost myelin by the circumference of the outermost myelin, which can circumvent the effect of myelin abnormality on the thickness counting to some extent. We mentioned it in the manuscript “In other words, the larger this ratio is, the thinner the myelin sheath is. The results showed that the ratio of myelin in the KO group of Mautner cells and other ventral axons was larger than that of the control group, i.e., the thickness of myelin was thinner, indicating immature myelination and possible partial loss of function”.

Point 3: The authors showed that the reduction of myelin and Jnk1 was upregulated in chd7-/-. However, Lorenzati et al. (2021) showed that the global ablation of the JNK1 results in a significant reduction of myelin in the cerebral cortex and corpus callosum at both postnatal and adult stages. Myelin alterations are accompanied by higher OPC density and proliferation during the first weeks of life, consistent with a transient alteration of mechanisms regulating OPC self-renewal and differentiation. Their data are opposite from the author's group. How do you explain the discrepancy?  Lorenzati M, Boda E, Parolisi R, Bonato M, Borsello T, Herdegen T, Buffo A, Vercelli A. c-Jun N-terminal kinase 1 (JNK1) modulates oligodendrocyte progenitor cell architecture, proliferation and myelination. Sci Rep. 2021 Mar 31;11(1):7264. doi: 10.1038/s41598-021-86673-6. PMID: 33790350; PMCID: PMC8012703.

Response 3: We thank the reviewer for pointing out the discrepancy. Previous studies have shown that c-Jun is a negative regulator of myelination[1] , LPS-sensitized HI causes white matter injury through JNK activation[2], FGF21 can act as a negative regulator for the myelin development process via activation of p38 MAPK/c-Jun[3]. Combined with these studies and our data, the phenomenon of increasing in phosphorylated JNK protein levels in the chd7-/- group is reasonable.

Point 4: The explanation of figure 7 a,b,c,d is not sufficient. The authors should provide more information to understand the figure in the main text. The font size is too small to read.

Response 3: We have changed the description of figure 7 a,b,c,d to “ a) Volcano plot showing each gene plotted according to its log2 fold change. All highly differentially expressed genes with P < 0.05 are in orange with fold change > 1.5. In total, 589 genes were significantly up-regulated and 438 genes were significantly down-regulated. b) Gene Ontology showing the 20 most significant functional classifications for differential gene enrichment. c) Gene Set Enrichment Analysis determining the significantly down-regulated pathway activity in the chd7 knockout group. d) Kyoto Encyclopedia of Genes and Genomes analysis determining the top 20 enriched pathways of the differentially expressed genes.”

The font size has also been revised to enhance readability.

Point 4: In the abstract, “We observed that knockout (KO) of chd7 intensely impeded the oligodendrocyte progenitor cells (OPCs) migration and myelin formation on account of massive expression of chd7 in OPCs, which might evoke upregulation of the MAPK signal pathway.” The authors made a global chd7-/- zebrafish. How "massive expression of chd7 in OPCs” happened?

Response 4: We accept the point and are grateful for the suggestion. In this revised version, we have changed this sentence to “We observed that knockout (KO) of chd7 intensely impeded the oligodendrocyte progenitor cells (OPCs) migration and myelin formation on account of massive expression of chd7 in oilg2+ cells”. We divided the Tg (olig2: EGFP) strain of zebrafish larvae into olig2+ cells and olig2- cells by flow sorting as the OLs were labeled by EGFP. The results showed that chd7 was highly expressed in oilg2+ cells.

Point 5: The supplemental data figure 2 is important data and should be in the main figure. 

Response 5: We thank for the valuable advice. We have included the supplemental figure 2 in the main figure 4.

Point 6: EM procedure in the method section is missing. How many myelinated axons were measured?  

Response 6: We thank the reviewer for the valuable comments. In this revised version, we have added EM procedure in method section and the content is as follows:

Freshly prepare MS-222 lethal working solution and glutaraldehyde fixative using PBS, anesthetize and lethally transfer the larvae to 2.5% glutaraldehyde (Sangon, China) quickly, and fix them overnight at room temperature on a shaker. The fixed fish were washed three times with 0.1 M Cacodylate buffer for 5 min each time. Fixed in 1% osmium acid for 2 hours at room temperature, and washed once with ddH2O. Gradient dehydration was performed using different concentrations of ethanol into 50% ethanol for 15 min before being placed in 70% uranyl acetate ethanol solution on a shaker overnight. Dehydration was continued using 80%, 95% and 100% ethanol for 15 min each. The dehydrated larvae were treated using a gradient of propylene oxide/epoxy resin for approximately 1 hour each time and finally embedded in epoxy resin. After orientation using pointed forceps, they were placed in an oven at 65°C for overnight curing. The cured resin blocks were briefly trimmed with a razor blade, placed in LKB NOVA ultra-thin sectioning machine (Lecia, Sweden) for ultra-thin sectioning (70-90 nm) and affixed to a copper mesh, and imaged using a JEM-1230 electron microscope (JEOL, Japan), and the remaining resin blocks were placed back into 2 mL centrifugal tubes and sealed in a drying box for storage. The acquired negatives were scanned with a scanner and analyzed and processed using ImageJ software.

In this study, 5 Mauthner cells and 102 myelinated axons cells of chd7+/+ group were measured while 6 Mauthner cells and 53 myelinated axons cells of chd7-/- group were measured.

Point 7: Marie et al. (2018) examined that chromatin remodeling by Chd7 and Chd8 requires OPC survival and differentiation. It should be included in the manuscript. Marie C, Clavairoly A, Frah M, Hmidan H, Yan J, Zhao C, Van Steenwinckel J, Daveau R, Zalc B, Hassan B, Thomas JL, Gressens P, Ravassard P, Moszer I, Martin DM, Lu QR, Parras C. Oligodendrocyte precursor survival and differentiation requires chromatin remodeling by Chd7 and Chd8. Proc Natl Acad Sci U S A. 2018 Aug 28;115(35): E8246-E8255. doi: 10.1073/pnas.1802620115. Epub 2018 Aug 14. PMID: 30108144; PMCID: PMC6126750.

Response 7: We thank the reviewer for the thoughtful comments. As suggested, we now cite this literature in our introduction: And it has been reported that oligodendrocyte precursor survival and differentiation require chromatin remodeling by Chd7 and Chd8 [24].

  1. Parkinson, D.B.; Bhaskaran, A.; Arthur-Farraj, P.; Noon, L.A.; Woodhoo, A.; Lloyd, A.C.; Feltri, M.L.; Wrabetz, L.; Behrens, A.; Mirsky, R.; et al. c-Jun is a negative regulator of myelination. The Journal of cell biology 2008, 181, 625-637, doi:10.1083/jcb.200803013.
  2. Wang, L.W.; Tu, Y.F.; Huang, C.C.; Ho, C.J. JNK signaling is the shared pathway linking neuroinflammation, blood-brain barrier disruption, and oligodendroglial apoptosis in the white matter injury of the immature brain. Journal of neuroinflammation 2012, 9, 175, doi:10.1186/1742-2094-9-175.
  3. Zhang, Y.; Jiang, K.; Xie, G.; Ding, J.; Peng, S.; Liu, X.; Sun, C.; Tang, X. FGF21 impedes peripheral myelin development by stimulating p38 MAPK/c-Jun axis. Journal of cellular physiology 2021, 236, 1345-1361, doi:10.1002/jcp.29942.
  4. Marie, C.; Clavairoly, A.; Frah, M.; Hmidan, H.; Yan, J.; Zhao, C.; Van Steenwinckel, J.; Daveau, R.; Zalc, B.; Hassan, B.; et al. Oligodendrocyte precursor survival and differentiation requires chromatin remodeling by Chd7 and Chd8. Proceedings of the National Academy of Sciences of the United States of America 2018, 115, E8246-e8255, doi:10.1073/pnas.1802620115.

Reviewer 2 Report

Authors were done great works to generate chd7 KO zebrafish and try to understand molecular mechanisms of Chd7 in oligodendrocytes using this fish. The study has potential to be excellent study after authors clear some issues as below. Also, the study should be reorganized based on previous mouse model.

1.     Since previous study has demonstrated Chd7 KO affected OPC proliferation and authors did not count total olig2-positive cells in figure 4, authors have to count total olig2-positive cells and evaluate if Chd7 depletion impair cell migration or cell proliferation. Also, Olig2-GFP transgenic zebrafish background is so high. Can authors count olig2-positive cells accurately?

2.     In spinal cord, olig2 positive cells are differentiated into oligodendrocytes, motor neurons, or astrocytes as you know. Because expression of isl1 and isl2 mRNA in chd7 KO zebrafish are higher compared with wild type. Thus, authors have to check if chd7 depletion facilitates motor neuron’s differentiation or not.

3.     EM data in Figure 6(a, b) looks no significant difference for myelination and quality of data is not good for publication.

4.     In Figure 7, authors were focusing on JNK and JNK phosphorylation in oligodendrocytes based on previous transcriptome sequence data of GABAergic neurons (Jamadagni et al., 2021). Because these cell’s signaling transduction pathway and reaction are totally big difference, author could not refer to previous study.

5.     Authors must discuss more about the association between oligodendrocyte maturation and JNK phosphorylation. 

The manuscript is written well. And quality of English is also high.

Author Response

Reviewer #2: Authors were done great works to generate chd7 KO zebrafish and try to understand molecular mechanisms of Chd7 in oligodendrocytes using this fish. The study has potential to be excellent study after authors clear some issues as below. Also, the study should be reorganized based on previous mouse model.

Answer:

We sincerely appreciate the constructive suggestions on our manuscript and the opportunity to revise it. We sincerely appreciate the constructive suggestions and the opportunity to revise our manuscript. CHD7 function in regulating myelin development on mice has indeed been reported in previous studies, however, generating zebrafish-based tools to study chd7 function may also be a useful addition to the field, considering that the chd7 gene is closely associated with CHARGE syndrome. In this study, we have obtained the conclusion that chd7 knockout hinders myelination in zebrafish oligodendrocytes.

 We hope the revised manuscript is suitable for possible publication.

Point 1: Since previous study has demonstrated Chd7 KO affected OPC proliferation and authors did not count total olig2-positive cells in figure 4, authors have to count total olig2-positive cells and evaluate if Chd7 depletion impair cell migration or cell proliferation. Also, Olig2-GFP transgenic zebrafish background is so high. Can authors count olig2-positive cells accurately?

Response 1: We accept the point and are grateful for the suggestion. Supplementary figure 2 shows that there was no significant change in the total number of oilg2+ cells after the deletion of the chd7 gene. We will adjust the microscope parameters to minimize background interference as much as possible, and only those that are significantly brighter than the background and meet the standard size and shape will be included in the cell count. In addition, our microscope can achieve stereo scanning, and we will view images of each z-layer to help count olig2+ cells.

Point 2: In spinal cord, olig2 positive cells are differentiated into oligodendrocytes, motor neurons, or astrocytes as you know. Because expression of isl1 and isl2 mRNA in chd7 KO zebrafish are higher compared with wild type. Thus, authors have to check if chd7 depletion facilitates motor neuron’s differentiation or not.

Response 2: This is an excellent point to make the paper complete, and we appreciate it. We found that the chd7-/- group swam longer distances for the same durations compared to the control group, after which the expression of motor neuron marker genes was measured. The upregulated isl1 and isl2a were used as additional index that chd7 knockout causes hyperactivity in zebrafish.

Point 3: EM data in Figure 6 (a, b) looks no significant difference for myelination and quality of data is not good for publication.

Response3: We agree with the professional advice. As suggested, we replaced the electron microscopy images in the current manuscript.

Point 4: In Figure 7, authors were focusing on JNK and JNK phosphorylation in oligodendrocytes based on previous transcriptome sequence data of GABAergic neurons (Jamadagni et al., 2021). Because these cell’s signaling transduction pathway and reaction are totally big difference, author could not refer to previous study.

Response 4: The reviewer’s comments are really thoughtful and we appreciate it. For the RNA-seq data sources we utilized (GSE139623), the contributors performed an unbiased transcriptomic analysis on zebrafish chd7-/- larval brains compared with wild‐type controls. Since the sample was 5 dpf dissected larval brains, it is reasonable to expect that we examine the effect of chd7 on oligodendrocytes through this batch of data.

Point 5: Authors must discuss more about the association between oligodendrocyte maturation and JNK phosphorylation.

Response 5: We thank for the valuable advice. In the current manuscript, we added more description of JNK phosphorylation and myelination process and the content is as follows: Previous studies have shown that c-Jun is a negative regulator of myelination [36], LPS-sensitized HI causes white matter injury through JNK activation [37], FGF21 can act as a negative regulator for the myelin development process via activation of p38 MAPK/c-Jun [38].

  1. Parkinson, D.B.; Bhaskaran, A.; Arthur-Farraj, P.; Noon, L.A.; Woodhoo, A.; Lloyd, A.C.; Feltri, M.L.; Wrabetz, L.; Behrens, A.; Mirsky, R.; et al. c-Jun is a negative regulator of myelination. The Journal of cell biology 2008, 181, 625-637, doi:10.1083/jcb.200803013.
  2. Wang, L.W.; Tu, Y.F.; Huang, C.C.; Ho, C.J. JNK signaling is the shared pathway linking neuroinflammation, blood-brain barrier disruption, and oligodendroglial apoptosis in the white matter injury of the immature brain. Journal of neuroinflammation 2012, 9, 175, doi:10.1186/1742-2094-9-175.
  3. Zhang, Y.; Jiang, K.; Xie, G.; Ding, J.; Peng, S.; Liu, X.; Sun, C.; Tang, X. FGF21 impedes peripheral myelin development by stimulating p38 MAPK/c-Jun axis. Journal of cellular physiology 2021, 236, 1345-1361, doi:10.1002/jcp.29942.

Round 2

Reviewer 2 Report

The authors reported the number of motor neurons were increased because the expression levels of isl1 and isl2a mRNA were unregulated in chd7 KO fish. If possible, authors may have to observe and count motor neurons by immunostaining or ISH to support their results.

Although the number of total olig2 positive cells was not significant difference between WT and KO fish,  the number of migrating OPC was decreased in KO. These phenomena may affect myelination and a supportive data.